

# Association of epilepsy and asthma: a population-based retrospective cohort study

Kuo-Liang Chiang[1,2,3,*], Fang-Chuan Kuo[4], Jen-Yu Lee[5,*] and Chin-Yin Huang[6]

[1] Department of Pediatric Neurology, Kuang-Tien General Hospital, Taichung, Taiwan
[2] Department of Nutrition, Hungkuang University, Taichung, Taiwan
[3] Department of Industrial Engineering and Enterprise Information, Tunghai University, Taichung, Taiwan
[4] Department of Physical Therapy, Hungkuang University, Taichung, Taiwan
[5] Department of Statistics, Feng Chia University, Taichung, Taiwan
[6] Program for Health Administration, Tunghai University, Taichung, Taiwan
[*] These authors contributed equally to this work.

## ABSTRACT

**Background**. Epidemiologic data supporting the epilepsy–asthma association are insufficient. Therefore, we examined this association in this study.

**Methods**. By using claims data from the National Health Insurance Research Database (Taiwan), we executed a retrospective cohort analysis. Analysis 1 entailed comparing 150,827 patients diagnosed as having incident asthma during 1996–2013 with disease-free controls who were selected randomly during the same period, frequency matched in terms of age and sex. Similarly, analysis 2 entailed comparing 25,274 patients newly diagnosed as having epilepsy with sex- and age-matched controls who were selected randomly. At the end of 2013, we evaluated in analysis 1 the epilepsy incidence and risk and evaluated in analysis 2 the asthma incidence and risk. We applied Kaplan–Meier analysis to derive plots of the proportion of asthma-free seizures.

**Results**. In analysis 1, the asthma group exhibited a higher epilepsy incidence than did the control group (3.05 versus 2.26 per 1,000 person-years; adjusted hazard ratio: 1.39, 95% CI [1.33–1.45]). We also noted a greater risk of subsequent epilepsy in women and girls. In analysis 2, we determined that the asthma incidence between the control and epilepsy groups did not differ significantly; however, some age subgroups including children and individuals in their 30s had an increased risk. A negative association was found in adolescents. The Kaplan–Meier analysis revealed epilepsy to be positively associated with subsequent onset of asthma within seven years of epilepsy diagnosis.

**Discussion**. Asthma may be associated with high epilepsy risk, and epilepsy may be associated with high asthma risk among children and individuals in their 30s. Nevertheless, people with epilepsy in other age subgroups should be aware of the possibility of developing asthma within seven years of epilepsy diagnosis.

Corresponding author
Kuo-Liang Chiang,
lambier.tw@yahoo.com.tw

## INTRODUCTION

Asthma and epilepsy are syndromes of heterogeneous diseases with likely multifactorial origins. Epilepsy, a ubiquitous neurological disease, is characterized by the occurrence of convulsive and nonconvulsive seizures. Estimates have revealed that epilepsy affects more than 50 million people worldwide, including two million people in the United States (*Browne & Holmes, 2001*; *Chang & Lowenstein, 2003*), and Moreover, asthma, a ubiquitous chronic disease, inherently involves changing degrees of airflow obstruction and airway inflammation as well as coughing, wheezing, and dyspnea episodes. Asthma has been estimated to affect over 300 million people worldwide, including 20 million people in the United States (*Masoli et al., 2004*; *Olin & Wechsler, 2014*). In Taiwan, the age-adjusted prevalence of epilepsy was 5.85 (per 1000) between 2000 and 2003 (*Chen et al., 2012*); the period prevalence of asthma was 11.9% between 2000 and 2007 (*Hwang et al., 2010*). Asthma and epilepsy are common causes of acute hospitalization. In the United States, epilepsy or convulsions make up 1% of all visits to the emergency department (ED) annually, and epilepsy uniformly constitutes 3.6% of all hospital admissions and admission rates across the country (*Pallin et al., 2008*; *Stump, 2008*). In the United States, acute asthma constitutes approximately 2% of all ED visits and 1.3% of all annual hospitalizations (*Moorman et al., 2007*). The annual global burden due to 23.4 million individuals with epilepsy in 2015 was estimated to be 12.4 million disability-adjusted life years (DALYs) (*Feigin et al., 2017*); 358.2 million individuals with asthma caused 26.2 million DALYs (*Soriano et al., 2017*). Thus, epilepsy and asthma are cost-intensive, complicated global health concerns (*England et al., 2012*; *Pawankar, 2014*; *Nunes, Pereira & Morais-Almeida, 2017*) that tend to reduce patients' quality of life (*Wang et al., 2012*) and place a considerable burden on their family members.

Asthma is associated with multiple neuropsychiatric disorders in children and adults, some of which are anxiety, attention deficit hyperactivity disorder, autism, and depression (*Fasmer et al., 2011*; *Van Lieshout & Macqueen, 2012*; *Brumpton et al., 2013*; *Oh et al., 2015*; *Zheng et al., 2016*; *Choi, Kim & Lee, 2017*). However, the association of epilepsy with asthma has never been clarified. Questions such as whether epilepsy can manifest as a bronchial spasm in asthma pathogenesis and whether people with asthma are more likely to develop epilepsy remain to be resolved. Immunity and inflammation seem to play a role in both disorders, with a sufficient body of experimental and clinical evidence suggesting that brain inflammation has the likelihood of predisposing a patient to and precipitating and perpetuating epileptogenesis. Certain systemic and neurological autoimmune disorders are reportedly affiliated with epilepsy, such as lupus cerebritis, Hashimoto's encephalopathy, multiple sclerosis, Rasmussen's encephalitis, linear scleroderma, and anti-N-methyl-D-aspartate receptor encephalitis (*Chiang et al., 2009a*; *Chiang et al., 2009b*; *Vincent & Crino, 2011*; *Devinsky, Schein & Najjar, 2013*). Proinflammatory cytokines play neuromodulatory functions, in addition to contributing to aberrant neuronal excitability underlying seizure disorders. Epilepsy and asthma share excitatory factors such as voltage-gated sodium channels, glutamate (i.e., N-methyl-D-aspartic acid), and acetylcholine, as well as

inhibitory factors such as voltage-gated potassium channels, γ-aminobutyric acid (gamma-aminobutyric acid [GABA]), glycine, and taurine. One study proposed a pathogenetic mechanism of asthma, similar to that of epilepsy, as a syndrome pertaining to genetically predisposed or inducible membrane hyperexcitability (*Hoang et al., 2006*). These findings imply a complex association between the two disorders.

Researchers have attempted for numerous decades to link epilepsy to asthma. Studies published prior to 1970 (*Levin, 1931*; *Forman, 1934*; *Speer, 1967*; *Fein & Kamin, 1968*) have hypothesized patients with asthma and other allergy conditions to be highly likely to experience seizures (and vice versa). The mentioned studies have applied terms that include "neurologic allergy" and "allergic epilepsy" to report allergies, concurrently presenting asthma, and epilepsy. Furthermore, some studies executed after 1970 have established a link between asthma and epilepsy based on electroencephalographic findings (*Mysik & Czerniawska-Mysik, 1972*; *Cinca & Dimitriu, 1976*; *Abramson, 1978*). However, despite growing bodies of evidence revealing such an association, only a few epidemiologic studies have been conducted on this association (*Castaneda et al., 1998*; *Kobau et al., 2004*; *Strine et al., 2005*; *Bilan & Ghaffari, 2008*; *Elliott, Moore & Lu, 2008*; *Silverberg, Joks & Durkin, 2014*), all of which had small sample sizes or used a cross-sectional design. Our review of the relevant literature showed that studies have yet to focus on this problem in Asian countries. In summary, large longitudinal studies in various populations are required to definitively explore the association between these two disorders and fill the knowledge gap.

In 1995, Taiwan implemented its National Health Insurance (NHI) program, which provides coverage for affordable, equitable, and universal health care to over 98% of Taiwan's 26 million residents (*Chien et al., 2012*). Similar to a natural laboratory setting, the NHI program's comprehensive health care coverage and continuous observation provide a special opportunity for the long-term observation of the correlation between epilepsy and asthma. Accordingly, in the present study, we utilized a longitudinal dataset from the National Health Insurance Research Database (NHIRD) and employed a large sample and representative data to execute a two-way population-based retrospective analysis in order to evaluate the asthma risk among patients with epilepsy as well as the epilepsy risk among patients with asthma.

## MATERIALS AND METHODS

### Data source

The data we used were subset data of the NHIRD, a database of longitudinal claims data covering a cohort of 1 million people randomly selected from among all of the insured NHI beneficiaries. Annually, the National Health Insurance Administration (NHIA) compiles NHI data and then arranges the data as data files (e.g., registration data and original claims data), to process reimbursements. For privacy protection, the files are deidentified through the scrambling of patient and medical facility identification codes and are subsequently dispatched to the National Health Research Institutes (NHRI), thus forming the original data files of the NHIRD.

We employed three types of data files–namely the ambulatory care claims, inpatient claims, and registry of beneficiaries–that are linked using an encrypted yet distinctive

personal identification number. The files contain patient data regarding medical history and demographic characteristics. Identifiers for individuals and medical facilities are unique to the NHIRD and researchers, and they cannot be applied to trace patients or providers of health services. Therefore, the requirement for full review was waived by the Institutional Review Board of Kuang Tien General Hospital, and the use of these data was authorized by the NHRI.

## Study design and patients

We evaluated the bidirectional epilepsy–asthma association by performing two analyses. In both analyses, we used the same procedures for selecting the patients (Fig. 1). Specifically, for analysis 1, we discovered patients with asthma (International Classification of Diseases, Ninth Revision, Clinical Modification (ICD-9-CM) code 493) (i.e., the asthma group) and established a control group comprising individuals without asthma. For analysis 2, we discovered patients with epilepsy (ICD-9-CM code 345) (i.e., the epilepsy group) and established a control group comprising individuals without epilepsy. In analyses 1 and 2, we assigned to the patient groups patients who had had inpatient claim or an initial ambulatory claim for asthma (analysis 1) or epilepsy (analysis 2) during 1996–2013. For each patient, the diagnosis date was defined to be the index date. The control groups were matched with the asthma (analysis 1) and epilepsy (analysis 2) groups during the study period. We applied a 1:1 control-to-case ratio. To investigate the temporal association of asthma with the subsequent occurrence of epilepsy, we excluded from analysis 1 patients who had been diagnosed as having epilepsy before receiving a diagnosis of asthma. Similarly, we excluded from analysis 2 patients who had been diagnosed as having asthma before epilepsy. We followed up all patients to determine the epilepsy or asthma incidence (analysis 1 and 2, respectively) until December 31, 2013, or their withdrawal from NHI coverage.

## Statistical analysis

R software (version 3.25; *R Core Team, 2016*) was used for all statistical analyses. In both analyses, we compared the differences in the categorical and continuous variables in the demographic data of the controls and patients by employing the chi-squared test and the independent-sample *t* test, respectively. Sex, age, household income, and residence location were analyzed. We stratified age into eight bands: 0–10, 11–20, 21–30, 31–40, 41–50, 51–60, 61–70, and >70 years. In addition, studies have reported that low socioeconomic status and low incomes are key risk factors not only for epilepsy (*Heaney et al., 2002*; *Kaiboriboon et al., 2013*) but also for the prevalence of asthma and increasing hospitalization due to asthma (*Ray et al., 1998*; *Markus, Lyon & Rosenbaum, 2010*). Thus, we employed household income as a confounding variable. We divided household income levels into four categories in terms of insurance premiums: >40,000NT$, 20,000–39,999NT$, 1–19,999NT$, and dependents. The NHIA and government consider individuals who do not receive a salary, such as unemployed people, students, and elderly people, as dependents. Similar to previous studies using NHIRD data, we classified residence location into the following regions of Taiwan: northern, central, southern, and eastern (*Chie et al., 1995*; *Lu et al., 2006*; *Chen et al., 2012*). Geographical variation, a demographic factor, refers to differences among populations of
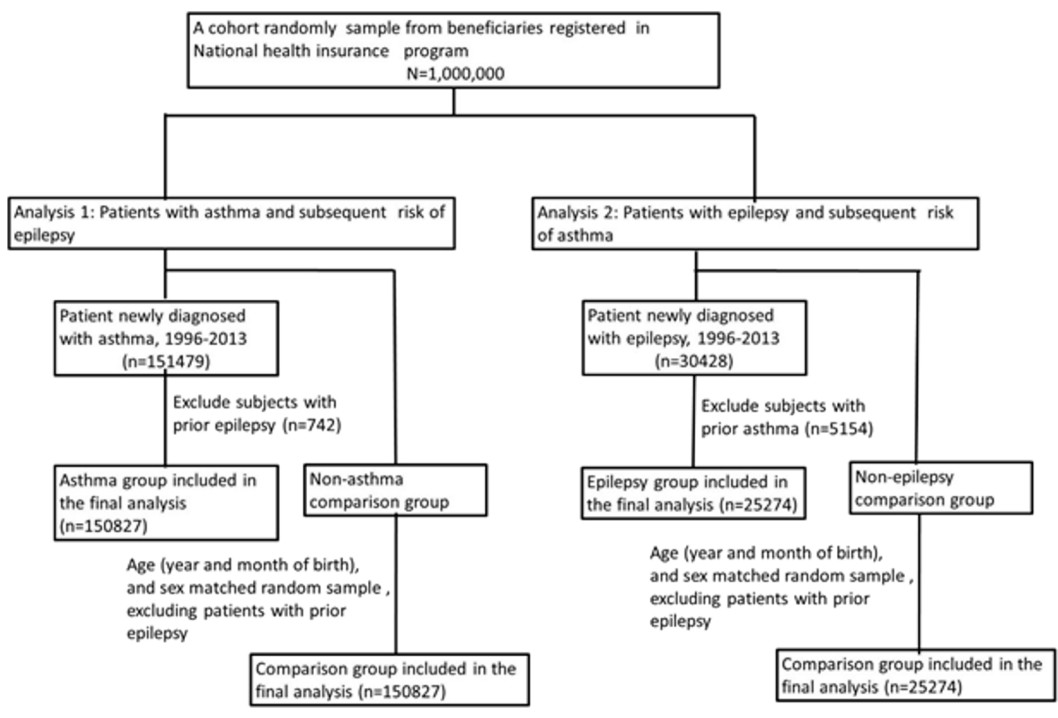

**Figure 1  Flowchart of study patient selection.**

a species in terms of genetic-based traits across the natural geographic range of the species. Compared with other regions, northern Taiwan comprises a higher number of economically and politically important cities, whereas eastern Taiwan comprises less politico-economic importance and a different demographic structure, with more indigenous people and a lower population density. One study reported geographic variation in epidemiologic patterns of epilepsy within Taiwan; eastern Taiwan had significantly higher prevalence and incidence of epilepsy than did other areas (*Chen et al., 2012*). Geographic variation also reported epidemiologic patterns of asthma in other countries (*Carvajal-Uruena et al., 2005*; *Malhotra et al., 2014*).

We assessed in analysis 1 the overall as well as sex-specific epilepsy incidence for patients with asthma and their corresponding controls. By applying Cox proportional hazard regression models, we estimated the hazard ratios (HRs) associated with epilepsy development in patients with asthma relative to that in the control group. Geographical variation and household income were adjusted for in the multivariate models. In addition, we applied Kaplan–Meier analysis to derive plots of the proportion of seizure-free asthma. We used the same data analysis procedures in both analyses. In analysis 2, we applied similar Cox regression models to evaluate the HRs associated with asthma incidence in patients with epilepsy relative to that in the control group. For both sets of analyses, we set statistical significance at ≤0.05. Additionally, we employed Kaplan–Meier analysis to derive plots of the proportion of asthma-free seizures.
**Table 1** Demographic characteristics of patients with asthma and controls.

| | Nonasthma comparison group | | Patients with asthma | | *p*-value |
|---|---|---|---|---|---|
| | *n* = 150,827 | Percentile | *n* = 150,827 | Percentile | |
| Characteristic | | | | | |
| Age (years) | | | | | |
| 0–10 | 47,394 | 31.42% | 47,394 | 31.42% | |
| 11–20 | 11,885 | 7.88% | 11,885 | 7.88% | |
| 21–30 | 13,044 | 8.65% | 13,044 | 8.65% | |
| 31–40 | 16,530 | 10.96% | 16,530 | 10.96% | |
| 41–50 | 17,076 | 11.32% | 17,076 | 11.32% | |
| 51–60 | 16,214 | 10.75% | 16,214 | 10.75% | |
| 61–70 | 14,640 | 9.71% | 14,640 | 9.71% | |
| >70 | 14,034 | 9.30% | 14,034 | 9.30% | |
| Mean (SD) of age | | 33.41 (25.51) | | 33.41 (25.51) | 1.00[*] |
| Sex | | | | | |
| Female | 76,998 | 51.05% | 76,998 | 51.05% | |
| Male | 73,819 | 48.94% | 73,819 | 48.94% | 1.00[**] |
| Income | | | | | |
| 0 | 69,082 | 45.80% | 70,137 | 46.50% | |
| <20,000 | 28,711 | 19.04% | 27,032 | 17.92% | |
| 20,000–39,999 | 38,688 | 25.65% | 39,128 | 25.94% | |
| >40,000 | 14,336 | 9.50% | 14,520 | 9.63% | <1.964e−13[*] |
| Residence location | | | | | |
| North | 68,234 | 45.24% | 78,449 | 52.01% | |
| Central | 37,455 | 24.83% | 35,953 | 23.84% | |
| South | 40,592 | 26.91% | 31,785 | 21.07% | |
| East | 3,505 | 2.32% | 3,715 | 2.46% | |
| Off island | 1,031 | 0.68% | 915 | 0.61% | <2.2e−16[*] |

**Notes.**
[*]*p*-value was calculated by Chi-square statistic.
[**]*p*-value was calculated by t statistic.

## RESULTS

### Asthma and risk of developing epilepsy

In the 1996–2013 data, we discovered 150,827 patients with asthma and 150,827 age- and sex-matched controls (Table 1); among these patients, 31.42% were aged 0–10 years and 51.05% were female patients. Household income data were similar between the two groups. Patients with asthma were indicated to be more likely to reside or work in northern Taiwan, compared with the controls (52.01% versus 45.24%). We observed the median follow-up periods in the asthma and control groups to be 5.8 and 6.2 years, respectively.

The asthma group exhibited a higher epilepsy incidence (3.05 versus 2.26 per 1,000 person-years; see Table 2). As revealed by the regression model, patients with asthma were 1.39 times more likely to develop epilepsy (95% CI [1.33–1.45]) after age, sex, income, and residence location had been controlled for. In both groups, the epilepsy incidence was higher in boys and men than it was in girls and women (Table 2). An association between

**Table 2** HRs for incidence of epilepsy in relation to asthma.

| | No asthma comparison Group | | | Patient with asthma | | | Hazard ratio and 95% CI (patient with asthma vs. comparison group) | |
|---|---|---|---|---|---|---|---|---|
| | Cases | PY | Incidence[a] | Cases | PY | Incidence[a] | Unadjusted | Adjusted[b] |
| ALL | 3,264 | 144,6014 | 2.26 | 4,236 | 138,6893 | 3.05 | 1.35* (1.29–1.41) | 1.39* (1.33–1.45) |
| **Stratified by SEX** | | | | | | | | |
| Female | 1,482 | 726,276 | 2.04 | 2,079 | 694,427 | 2.99 | 1.47* (1.37–1.57) | 1.52* (1.42–1.64) |
| Male | 1,782 | 719,738 | 2.48 | 2,157 | 692,466 | 3.12 | 1.25* (1.18–1.33) | 1.28* (1.21–1.37) |
| **Stratified by AGE** | | | | | | | | |
| 0–10 | 878 | 471,260 | 1.86 | 1,141 | 488,971 | 2.33 | 1.27* (1.16–1.38) | – |
| 11–20 | 184 | 125,089 | 1.47 | 178 | 113,217 | 1.57 | 1.05 (0.85–1.29) | – |
| 21–30 | 140 | 112,963 | 1.24 | 173 | 111,653 | 1.55 | 1.26* (1.01–1.57) | – |
| 31–40 | 191 | 143,218 | 1.33 | 296 | 140,908 | 2.10 | 1.57* (1.31–1.89) | – |
| 41–50 | 249 | 153,727 | 1.62 | 418 | 151,436 | 2.76 | 1.71* (1.46–1.99) | – |
| 51–60 | 359 | 148,633 | 2.42 | 495 | 134,844 | 3.67 | 1.52* (1.33–1.74) | – |
| 61–70 | 541 | 142,983 | 3.78 | 732 | 130,915 | 5.59 | 1.48* (1.33–1.66) | – |
| >70 | 722 | 148,142 | 4.88 | 803 | 114,948 | 6.99 | 1.39* (1.26–1.54) | – |
| **Stratified by INCOME** | | | | | | | | |
| 0 | 1,614 | 677,069 | 2.38 | 1,981 | 660,961 | 3.00 | 1.25* (1.17–1.33) | – |
| <20,000 | 784 | 278,760 | 2.81 | 1,011 | 253,786 | 3.98 | 1.41* (1.28–1.55) | – |
| 20,000–39,999 | 744 | 362,018 | 2.06 | 1,073 | 347,930 | 3.08 | 1.50* (1.37–1.65) | – |
| >40,000 | 122 | 128,167 | 0.95 | 171 | 124,216 | 1.38 | 1.46* (1.15–1.84) | – |
| **Stratified by LOCATION** | | | | | | | | |
| N | 1,385 | 650,132 | 2.13 | 1,885 | 713,899 | 2.64 | 1.23* (1.15–1.32) | – |
| C | 958 | 360,406 | 2.66 | 1,230 | 335,659 | 3.66 | 1.38* (1.27–1.50) | – |
| S | 802 | 391,822 | 2.05 | 953 | 295,787 | 3.22 | 1.57* (1.43–1.73) | – |
| E | 97 | 33,574 | 2.89 | 139 | 33,990 | 4.09 | 1.41* (1.08–1.82) | – |
| O | 22 | 10,081 | 2.18 | 29 | 7,558 | 3.84 | 1.71 (0.98–2.97) | – |

**Notes.**
PY: person-years at risk.
[a]The incidence is per 1,000 person-year.
[b]Adjusted for age, sex, income and residence location.
*Significance at 0.05 level.

asthma and increased epilepsy incidence was observed in both sexes, with a significantly higher relative risk estimate ($t$ test) in girls and women (adjusted HR: 1.52, 95% CI [1.42–1.64]) than in boys and men (adjusted HR: 1.28, 95% CI [1.21–1.37]). In addition, the epilepsy and control groups both had lower subsequent incidence of asthma in patients among the highest household income bracket (>NT$40,000); among patients who lived in eastern Taiwan, both groups also had higher subsequent incidence of asthma.

The Kaplan–Meier analysis results revealed a consistent positive association between asthma and subsequent epilepsy development (Fig. 2).

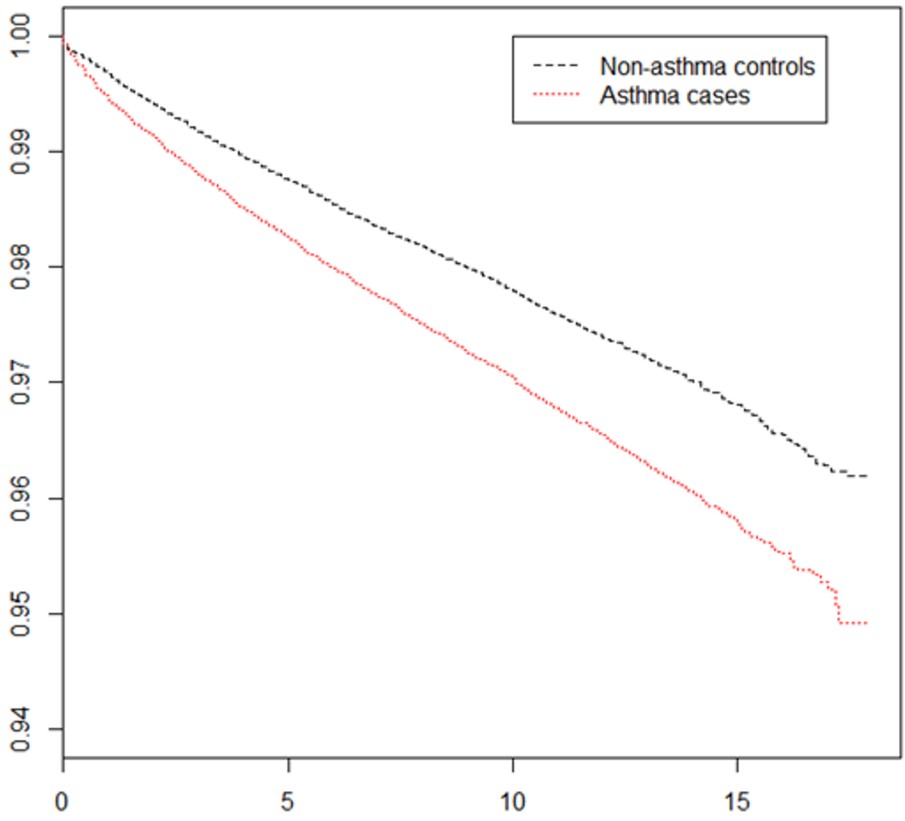

**Figure 2** Survival curve of epilepsy development among patients with asthma and the control group.

## Epilepsy and corresponding risk of asthma

In the 1996–2013 data, we identified 25,274 patients with epilepsy and 25,274 age-matched and sex-matched controls (Table 3); among these patients, 20.3% were aged <10 years, 14.2% were aged >70 years, and 9.2%–12.8% were in each of the other age groups, and 45.5% were also female patients. We observed epilepsy group to be more likely to have a lower household income, compared with the control group (dependent–lowest income level: 18.9% versus 24.3%). Moreover, patients with epilepsy were found to be more likely to reside or work in northern Taiwan than were their controls (45.3% versus 42.8%). We noted the median follow-up periods in the epilepsy and control groups to be 6.1 and 6.4 years, respectively.

Although the epilepsy group exhibited a higher asthma incidence than did the control group (18.79 versus 17.75 per 1,000 person-years; Table 4), the regression model did not provide supportive evidence (adjusted HR: 0.96, 95% CI [0.91–0.999]) after age, sex, income, and residence location had been controlled for. Although the results indicated a significant negative association, only mild influence was observed. A similar result was observed for male patients with epilepsy versus male individuals without epilepsy (adjusted HR: 0.91, 95% CI [0.86–0.97]). However, we observed a positive association in some age subgroups, including people aged 0–10 and 31–40 years. In the 0–10-year subgroup, the

**Table 3 Demographic characteristics of patients with epilepsy and controls.**

| | Nonepilepsy comparison group | | Patients with epilepsy | | p-value |
|---|---|---|---|---|---|
| | n = 25,274 | Percentile | n = 25,274 | Percentile | |
| Characteristic age (years) | | | | | |
| 0–10 | 5,122 | 20.27% | 5,122 | 20.27% | |
| 11–20 | 2,326 | 9.20% | 2,326 | 9.20% | |
| 21–30 | 2,515 | 9.95% | 2,515 | 9.95% | |
| 31–40 | 2,798 | 11.07% | 2,798 | 11.07% | |
| 41–50 | 3,231 | 12.78% | 3,231 | 12.78% | |
| 51–60 | 3,010 | 11.91% | 3,010 | 11.91% | |
| 61–70 | 2,688 | 10.64% | 2,688 | 10.64% | |
| >70 | 3,584 | 14.18% | 3,584 | 14.18% | |
| Mean (SD) of age | | 39.13 (25.92) | | 39.13 (25.92) | 1.00[*] |
| Sex | | | | | |
| Female | 11,499 | 45.50% | 11,499 | 45.50% | |
| Male | 13,775 | 54.50% | 13,775 | 54.50% | 1.00[**] |
| Income | | | | | |
| 0 | 9,715 | 38.44% | 10,313 | 40.80% | |
| <20,000 | 5,215 | 20.63% | 6,707 | 26.54% | |
| 20,000–39,999 | 7,469 | 29.55% | 6,597 | 26.10% | |
| >40,000 | 2,875 | 11.38% | 1,657 | 6.56% | <2.2e−16[*] |
| Residence location | | | | | |
| North | 10,815 | 42.79% | 11,447 | 45.29% | |
| Central | 7,148 | 28.28% | 6,329 | 25.04% | |
| South | 6,306 | 24.95% | 6,665 | 26.37% | |
| East | 818 | 3.24% | 644 | 2.55% | |
| Off island | 187 | 0.74% | 189 | 0.75% | <2.2e−16[*] |

**Notes.**
[*]p-value was calculated by Chi-square statistic.
[**]p-value was calculated by t statistic.

epilepsy group exhibited a higher asthma incidence than did the control group (36.43 versus 33.85 per 1,000 person-years; adjusted HR: 1.09, 95% CI [1.01–1.16]). In the 31–40-year subgroup, the epilepsy group exhibited a higher asthma incidence than did the control group (10.73 versus 7.81 per 1,000 person-years; adjusted HR: 1.34, 95% CI [1.12–1.61]). By contrast, in the 11–20-year subgroup, the epilepsy group exhibited a lower asthma incidence than did the control group (7.92 versus 12.59 per 1,000 person-years; adjusted HR: 0.62, 95% CI [0.51–0.75]). In the >70-year subgroup, the epilepsy group exhibited a lower asthma incidence than did the control group (21.36 versus 24.46 per 1,000 person-years; adjusted HR: 0.78, 95% CI [0.69–0.87]). In addition, the asthma and control groups both had lower subsequent incidence of epilepsy among patients in the highest household income bracket; among patients who lived in southern Taiwan, both groups also had lower subsequent incidence of epilepsy.

**Table 4  HRs for incidence of asthma in relation to epilepsy.**

| | No epilepsy comparison Group | | | Patient with epilepsy | | | Hazard ratio and 95% CI (patient with Epilepsy vs. comparison group) | |
|---|---|---|---|---|---|---|---|---|
| | Cases | PY | Incidence[a] | Cases | PY | Incidence[a] | Unadjusted | Adjusted[b] |
| **ALL** | 4,235 | 238,577 | 17.75 | 3,752 | 199,639 | 18.79 | 1.00 (0.96–1.04) | 0.96[*] (0.91–0.999) |
| **Stratified by SEX** | | | | | | | | |
| Female | 1,866 | 106,319 | 17.55 | 1,712 | 90,654 | 18.89 | 1.03 (0.96–1.10) | 1.02 (0.95–1. 08) |
| Male | 2,369 | 132,259 | 17.91 | 2,040 | 108,984 | 18.72 | 0.97 (0.92–1.03) | 0.91[*] (0.86–0.97) |
| **Stratified by AGE** | | | | | | | | |
| 0–10 | 1,561 | 46,110 | 33.85 | 1,691 | 46,423 | 36.43 | 1.09[*] (1.01–1.16) | – |
| 11–20 | 287 | 22,799 | 12.59 | 174 | 21,970 | 7.92 | 0.62[*] (0.51–0.75) | – |
| 21–30 | 206 | 25,423 | 8.10 | 193 | 22,582 | 8.55 | 1.04 (0.86–1.27) | – |
| 31–40 | 216 | 27,674 | 7.81 | 255 | 23,773 | 10.73 | 1.34[*] (1.12–1.61) | – |
| 41–50 | 325 | 30,930 | 10.51 | 305 | 25,436 | 11.99 | 1.11 (0.94–1.29) | – |
| 51–60 | 379 | 27,604 | 13.73 | 301 | 20,265 | 14.85 | 1.05 (0.90–1.22) | – |
| 61–70 | 446 | 24,722 | 18.04 | 385 | 18,213 | 21.14 | 1.12 (0.98–1.29) | – |
| >70 | 815 | 33,316 | 24.46 | 448 | 20,976 | 21.36 | 0.78[*] (0.69–0.87) | – |
| **Stratified by INCOME** | | | | | | | | |
| 0 | 2,256 | 87,760 | 25.71 | 2,163 | 80,812 | 26.77 | 0.98(093.–1.04) | – |
| <20,000 | 796 | 51,479 | 15.46 | 811 | 54,368 | 14.92 | 0.91(0.82–1..00) | – |
| 20,000–39,999 | 943 | 72,524 | 13.00 | 665 | 51,268 | 12.97 | 0. 97(0.88–1.07) | – |
| >40,000 | 258 | 26,858 | 9.61 | 132 | 13,241 | 9.97 | 1.02(0.825–1.26) | – |
| **Stratified by LOCATION** | | | | | | | | |
| N | 2,082 | 106,545 | 19.54 | 1,815 | 86,230 | 21.05 | 1.02 (0.96–1.09) | – |
| C | 1,047 | 59,993 | 17.45 | 1,056 | 55,292 | 19.10 | 1.03 (0.95–1.12) | – |
| S | 958 | 64,373 | 14.88 | 744 | 49,795 | 14.94 | 0.94 (0.85–1.03) | – |
| E | 113 | 5,852 | 19.31 | 113 | 6,660 | 16.97 | 0.85 (0.66–1.11) | – |
| O | 35 | 1,815 | 19.29 | 24 | 1,662 | 14.44 | 0.74 (0.44–1.25) | – |

**Notes.**

PY: person-years at risk.

[a]The incidence is per 1,000 person-year.

[b]Adjusted for age, sex, income and residence location.

[c]Significance at 0.05 level.

The Kaplan–Meier analysis revealed epilepsy to be positively associated with subsequent development of asthma within the first seven years of epilepsy diagnosis. After seven years, the results indicated a negative association (Fig. 3).

## DISCUSSION

Our review of the relevant literature indicated the current study to be the first study to apply a longitudinal dataset to execute a two-way population-based retrospective analysis in order to evaluate the asthma risk among patients with epilepsy as well epilepsy risk among patients with asthma. Our results confirm our hypothesis stating that patients with asthma are more likely to develop epilepsy than are other individuals. In addition, female patients with asthma had a significantly higher risk of epilepsy than did male patients with

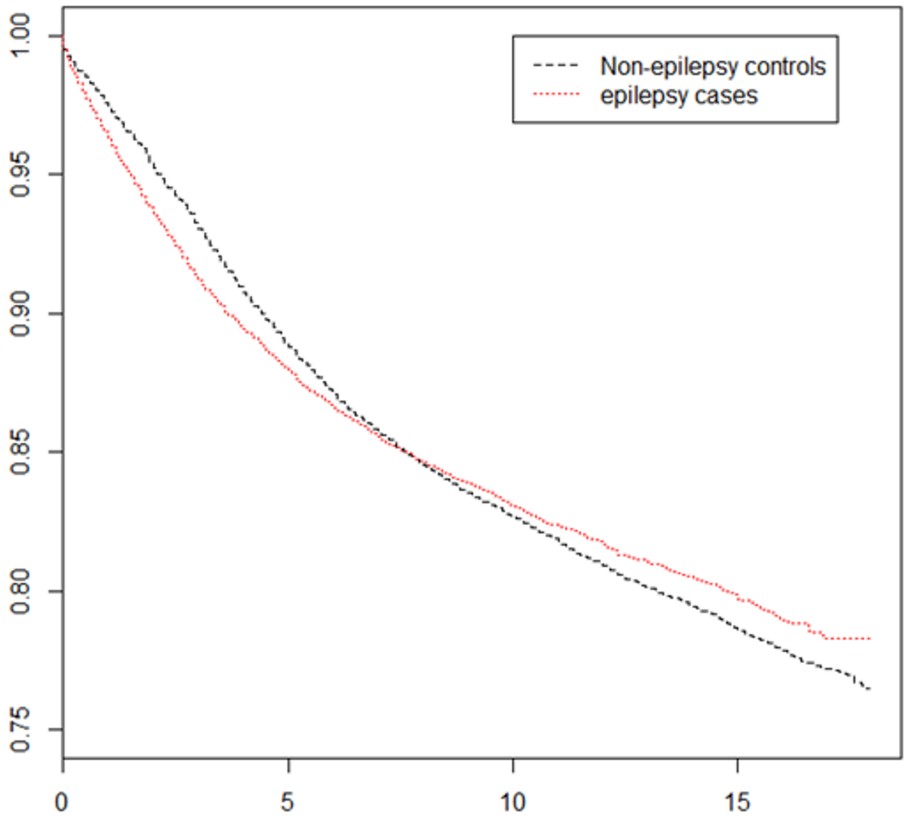

**Figure 3** Survival curve of asthma development among patients with epilepsy and the control group.

asthma. By contrast, patients with epilepsy did not have a significantly increased risk of asthma compared with individuals without epilepsy. However, the Kaplan–Meier analysis revealed epilepsy to be positively associated with subsequent asthma development within the first seven years of epilepsy diagnosis. Furthermore, patients diagnosed as having epilepsy at the ages of 0–10 and 31–40 years were indicated to be at an increased risk of asthma in later life, whereas those diagnosed at the ages of 11–20 and >70 years were indicated to be at a decreased risk of asthma in later life, compared with other individuals. Similar to the results of previous studies, the present study revealed that household income and geographic variation influenced incidence of asthma and epilepsy. Eastern Taiwan had a higher incidence of epilepsy than did other areas, and southern Taiwan had a lower incidence of asthma than did other areas. In addition, the population in the highest household income bracket had lower incidences of epilepsy and asthma than did those in all lower brackets.

This study observed a significant temporal association of asthma with epilepsy risk. Some clinical cases of epilepsy induced by long-standing asthma support our results (*Costello & Fox, 1936*; *Czubkowska et al., 1994*). A study involving the surveillance of epilepsy in 800 children with asthma reported that 26 children subsequently developed epilepsy and that asthmatic children exhibited a significantly higher epilepsy prevalence rate than did the

general population (*Bilan & Ghaffari, 2008*). A US population–based study (*Silverberg, Joks & Durkin, 2014*) that applied the 2007–2008 National Survey of Children's Health showed that children having ≥1 allergy disease exhibited a higher frequency of epileptic episodes in their lifetime than did their nonallergic counterparts.

Our results revealed that patients with asthma are more prone to epilepsy, but not vice versa. The association between the two disorders may be primarily the result of the sequelae of asthma diagnosis. The most likely mechanism linking asthma to epilepsy development is anoxia due to frequent asthma attacks; however, we were unable to address this mechanism because the frequencies of asthma attacks and epileptic seizures could not be determine based on the current dataset. Another direct mechanism is hypocapnia. Hypocapnia, caused by hyperventilation, is commonly observed in patients with asthma (*Bruton & Holgate, 2005*), and it can induce vasoconstriction causing cerebral hypoperfusion, diminishing oxygen delivery, and increasing neuronal excitation by releasing glutamate and dopamine. Hyperventilation engenders interictal discharges, which are particularly relevant in idiopathic seizures. Hyperventilation-induced activation is significantly more frequent in the temporal lobe than in other lobes (*Guaranha et al., 2005*). Furthermore, hyperventilation induces absence seizures (*Wirrell et al., 1996*). Apart from hypocapnia, the most common finding in terms of acid–base disturbances in patients with acute asthma is respiratory alkalosis (*Mountain et al., 1990*). The hypoxic effect of hypocapnia can exaggerate respiratory alkalosis, resulting in a left shift of the hemoglobin–oxygen dissociation curve. Furthermore, metabolic acidosis is prevalent in patients having severe asthma. The ultimate outcomes of the aforementioned processes include reduced cerebral blood flow and oxygen transport and delivery to the brain. Additionally, alkalosis was reported to promote calcium binding to plasma albumin, resulting in a reduced serum level of ionized calcium (*Locatto et al., 1984*), which can increase the probability of seizures. Adding xanthine-containing antiasthmatic agents such as theophylline and aminophylline with potential central nervous system (CNS) stimulant properties may further complicate the management of epilepsy because of a lack of seizure control or precipitation of seizures in epileptic asthmatics. Theophylline-associated seizures are considered a neurological emergency because they can be intractable. Such seizures were reported in patients with low and therapeutic serum levels of theophylline, indicating that seizures are not merely a result of dose-dependent drug toxicity (*Dunn & Parekh, 1991*).

Our results revealed no significantly higher risk of asthma in patients with epilepsy than in individuals without epilepsy. Per our review of the relevant literature, no studies have determined whether epilepsy is a risk factor for asthma. However, we noted epilepsy to be positively associated with asthma development within the first seven years of epilepsy diagnosis. In age subgroup analysis, patients diagnosed as having epilepsy aged 0–10 or 31–40 years were observed to have a high risk of asthma development in later life, whereas those diagnosed as having epilepsy aged 11–20 or >70 years were observed to have a low risk of asthma development in later life. Therefore, we believe that the epilepsy–asthma association is complicated and multifactorial. A Taiwanese study reported an asthma risk in children who experience febrile seizures; the findings of the study partially support our findings in the 0–10-year subgroup (*Lin et al., 2014*).

Chronic inflammation is a crucial mechanism in both asthma and epilepsy. The epilepsy–asthma association may be explained by several hypotheses. First, the two disorders perhaps share similar susceptible genes, including de novo mutations and cytokine genes. Moreover, cytokines oversecreted following a course of atopic reactions can pass through the blood–brain barrier, subsequently contributing to neuroinflammation and damaging neuron cells. *Mao et al. (2013)* positively correlated interleukin (IL)-17a, IL-6, interferon (IFN)-γ, and IL-10 levels with the severity and frequency of seizures and suggested that the IL-17a level was correlated significantly with seizure frequency. Another study revealed the IL-6-174G>C (rs1800795) polymorphism to be significantly associated with the frequency of epileptic seizures and with the incidence of drug-refractory epilepsy (*Tiwari et al., 2012*). Elevated levels of IL-6 and IL-1b have been reported in patients with asthma, with the cytokines being particularly overexpressed in the asthmatic bronchial epithelium (*Ishioka, 1996*; *Thomas & Chhabra, 2003*; *Martinez-Nunez et al., 2014*); this thus signifies that these cytokines might play in role in mediating proinflammatory processes and airway hyperresponsiveness underlying asthmatic diseases. The chronic neuroinflammation and neuronal damage resulting from the penetration of cytokines through the blood–brain barrier raises patient susceptibility to epileptogenesis.

Second, specific viral infections and injury-associated proinflammatory cytokines can trigger asthma and epilepsy; fever-related inflammatory cytokines as well as bacterial, specific viral, and other pathogenic infections can also contribute to pathogenesis. Arteritis, ischemia, and infarction are the main pathological outcomes of severe viral or bacterial CNS diseases and may result in seizures (*Sander, 2004*). In addition, a mild infection may activate the IL-1 receptor–toll-like receptor pathway; however, damage-associated molecular patterns including high mobility group box 1 could engender such activation (*Van Vliet et al., 2017*). Specific viral infections, such as adenovirus, herpes simplex virus, human herpesvirus-6, influenza, and respiratory syncytial virus, were associated with febrile seizures (*Lin et al., 2014*). Rotavirus and norovirus enteritis are associated with benign convulsions in childhood (*Kim et al., 2016*; *Yorulmaz, Sert & Yilmaz, 2017*). Furthermore, brain inflammation can be induced by the cyclooxygenase-2-induced production of prostaglandins (*Fabene et al., 2008*; *Serrano et al., 2011*). Additional bodies of evidence indicate that mast cells and microglial cells as well as their interaction substantially influence neuroinflammation (*Chiou & Hsieh, 2008*; *Skaper, Facci & Giusti, 2014*).

Channelopathies may partially contribute to comorbidities related to epilepsy and asthma. Channelopathies refer to diseases engendered by ion channel defects that are caused by genetic or acquired factors; ion-channel-encoding gene mutations are the most common cause of such diseases. Such defects have been implicated in an extensive range of disorders and conditions, which include migraine, epilepsy, deafness, blindness, diabetes, cardiac arrhythmia, hypertension, asthma, cancer, and irritable bowel syndrome. Absence epilepsy is reportedly associated with mutations in several ion-channel-encoding genes. For example, CACNA1H gene variants, denoting the α1H pore-forming subunit of T-type calcium channels, were observed in a patient subset diagnosed as having childhood absence epilepsy (CAE) (*Chen et al., 2003*). Scholars have additionally implicated missense GABRA1, GABRA6, GABARB3, and GABARG2 mutations—genes encoding numerous

GABA-A receptor subunits–in CAE (*Gurba et al., 2012*; *Kim, 2014*; *Spillane, Kullmann & Hanna, 2016*). The onset of CAE is commonly in the age range of 4 to 10 years, peaking at 5–7 years. More than half of all patients with CAE enter remission in adolescence. Such characteristics of channelopathies could partially explain why in our cohort of patients with epilepsy, those who had experienced the onset of epilepsy while aged 0–10 years were indicated to be at a high risk of asthma development in later life, whereas those who had experienced it while aged 11–20 years were indicated to be a decreased risk.

Some antiepileptic drugs (AEDs) might play roles in preventing and reducing the frequency of asthma attacks. Studies have suggested that in addition to the suppression of inflammation and bronchodilation, the control of excitatory mechanisms including voltage-gated sodium channel and glutamate receptors present in the brain and lung tissue, adjunctive glycine, and GABA agonists could facilitate the development of highly effective and safe asthma prevention and treatment strategies (*Hoang et al., 2006*; *Hoang et al., 2010*). Phenytoin was proven effective against acute and paroxysmal asthma, with most phenytoin-treated patients exhibiting sustained benefits even after treatment discontinuation (*Shulman, 1942*; *Sayar & Polvan, 1968*; *Jain & Jain, 1991*). Carbamazepine and sodium valproate have also been reported as having high antiasthmatic activities (*Lomia et al., 2005*). Lidocaine, a local anesthetic used in status epilepticus, works primarily by blocking sodium channels and decreasing membrane excitability. It was also effective in treating patients with severe steroid-dependent asthma (*Slaton, Thomas & Mbathi, 2013*). Our previous research showed that the AED prescription rate among children with epilepsy was lower than that among adolescents with epilepsy in Taiwan (Fig. 4) (*Chiang & Cheng, 2014*), and this could partially explain why in our epilepsy cohort of patients with epilepsy, those who had experienced the onset of epilepsy while aged 0–10 years were indicated to at a high risk of asthma development in later life, whereas those who had experienced it while aged 11–20 years were indicated to be at a decreased risk.

The insular cortex might be critical in the epilepsy–asthma association. Dyspnea is a high-impact symptom of asthma. A study noted that the periaqueductal gray matter downregulated affect-related insular cortex activity during sensations of pain and dyspnea in patients with asthma. The implicated insular cortex and periaqueductal gray matter might present as a mechanism of neuronal habituation, thereby ameliorating the corresponding unpleasantness of affective dyspnea in such patients (*Von Leupoldt et al., 2009*). In another study, a group of patients with insular cortex lesions were indicated to experience reduced perceptual sensitivity to dyspnea unpleasantness, compared with the corresponding control group (*Schon et al., 2008*). By contrast, some case reports have demonstrated that in patients with insular cortex epilepsy, a seizure attack could clinically present as dyspnea and chest tightness during sleep, and this could be difficult to distinguish from asthma-associated dyspnea.

Our study involves certain limitations that warrant consideration. First, in the claims data, the medical coding accuracy may influence data validity; this is because only people who sought medical aid were enlisted. In addition, diagnoses were based on inpatient claims and ambulatory care claims; hence, some may have represented a tentative diagnosis, rather than a final diagnosis. However, epilepsy is stigmatized in Taiwan, and thus identifying

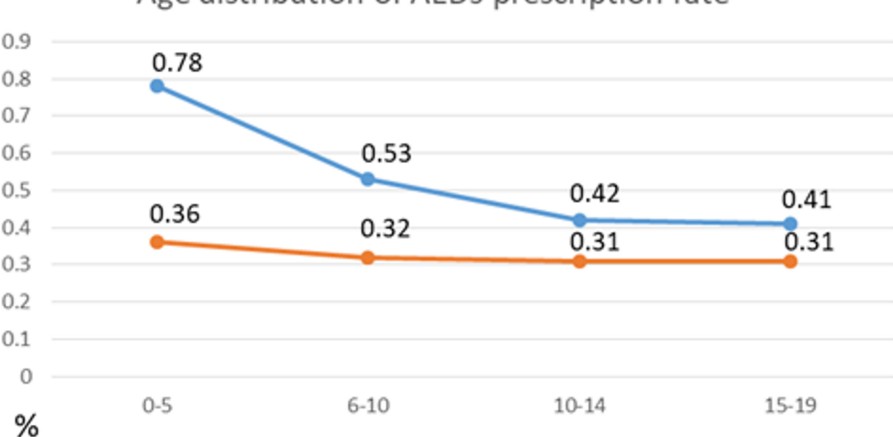

**Figure 4** Prevalence of epilepsy and epilepsy with AEDs prescription (data reference: *Chiang & Cheng, 2014*).

epilepsy in patients may be difficult in some cases. Although physicians apply the same diagnostic criteria, biases may be possible. However, the NHIA has a medical review system to monitor insurance claims; consequently, the effect of diagnostic validity on our findings is minimal. Second, the NHIRD limitations necessitated the use of only ICD-9-CM codes (i.e., 345 for epilepsy and 493 for asthma) for case patient identification. Further information such as syndromes, epilepsy classification, and asthma severity level was unavailable. Third, the NHIRD does not include medical record information on factors that might have exerted a minor influence on our results, such as the comorbidities of atopic dermatitis and allergic dermatitis, food allergies, family histories, personal lifestyles, and environmental factors. Fourth, the control group may have included patients with undiagnosed asthma or epilepsy. This sampling bias may be slightly overestimated in the control group but not sufficiently to affect the interpretation of the results. Finally, we did not analyze drug-related pathogenic or therapeutic effects such as the antiasthmatic effect of AEDs and epileptogenicity of xanthine-containing antiasthmatic agents.

## CONCLUSION

Despite the limitations in this study, our data suggest a complex epilepsy–asthma association. Patients with asthma had a higher risk of epilepsy than did individuals without asthma. By contrast, risk of asthma did not differ significantly between patients with epilepsy and individuals without epilepsy. However, patients experiencing epilepsy onset at specific ages had a higher or lower risk of asthma development than did individuals without epilepsy. The results revealed that patients with asthma are more prone to epilepsy, but not vice versa. One of the most likely explanations of the association between asthma and epilepsy is anoxia due to frequent asthma attacks or due to the side effects of antiasthma medication. To effectively fill the knowledge gap, future studies are recommended to

focus on incidence of epilepsy in patients with asthma of varying severity or in patients with varying levels of asthma control and compliance. Another possible explanation for the association between these two disorders is that they share common pathogenic mechanisms, which include hypocapnia, chronic inflammation, immune dysregulation, genetic susceptibility, channelopathies, drug-related pathogenic or therapeutic effects, and environmental factors. Additional studies should be conducted on these hypotheses to elucidate the mechanisms that link asthma, atopic dermatitis, or a broader atopy spectrum to epilepsy. In addition, our findings contributed to the literature on socioeconomic disparity, including the influence of household income and geographic variation on patients with epilepsy and those with asthma; physicians and public health policymakers should focus on this aspect. Reducing such disparity requires interventions to prevent epilepsy and asthma and improve the quality of care for these disorders.

## ACKNOWLEDGEMENTS

The authors thank the NHRI and the NHIA for providing the study data. This manuscript was edited by Wallace Academic Editing.

### Funding

The authors received no funding for this work.

### Competing Interests

The authors declare there are no competing interests.

### Author Contributions

- Kuo-Liang Chiang conceived and designed the experiments, analyzed the data, prepared figures and/or tables, approved the final draft.
- Fang-Chuan Kuo conceived and designed the experiments, authored or reviewed drafts of the paper.
- Jen-Yu Lee conceived and designed the experiments, performed the experiments, analyzed the data, contributed reagents/materials/analysis tools, prepared figures and/or tables.
- Chin-Yin Huang analyzed the data, contributed reagents/materials/analysis tools, authored or reviewed drafts of the paper.

### Ethics

The following information was supplied relating to ethical approvals (i.e., approving body and any reference numbers):

The requirement for full review was waived by the Institutional Review Board of Kuang Tien General Hospital, and the use of these data was authorized by the NHRI.
### Data Availability

Data are available from the National Health Insurance Research Database (NHIRD) published by Taiwan National Health Insurance (NHI) Bureau. The data utilized in this study cannot be made available in the paper, the supplemental files, or in a public repository due to the "Personal Information Protection Act" executed by Taiwan's government, starting from 2012. Requests for data can be sent as a formal proposal to the NHIRD (http://nhird.nhri.org.tw) or by email to nhird@nhri.org.tw.

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
