# Peer review of "Association of epilepsy and asthma: a population-based retrospective cohort study"

_PeerJ, doi:10.7717/peerj.4792_

## Round 0.1 · original submission · Minor Revisions

Please pay particular attention to the comments under the validity section by Reviewer 2. Thank you for submitting your work to PeerJ.

Reviewer 1 ·

Basic reporting

In lines 83 and 97 the authors use the phrase “causal relationship” in discussing the relationship between asthma and epilepsy. I would suggest rewording these two sentences since it is unclear if an association exists between the disorders and causation cannot be determined based on an epidemiologic study.

Experimental design

Did you consider additional analyses looking at epilepsy incidence in individuals with low vs. high asthma severity, or in those with uncontrolled vs. controlled asthma? This would be interesting to consider or to discuss as possible future directions.

Validity of the findings

1. Lines 181-184 of the results note that there was a nonsignificantly higher relative risk estimate in females vs. males; however, the confidence intervals for the estimates in females and males do not overlap, suggesting a statistically significant difference. Please clarify this and specify in the manuscript how significance was tested for the interaction with gender.
2. In lines 219-220, the statement that “patients with epilepsy did not have a significant risk of asthma” is unclear. This would be clearer as “patients with epilepsy did not have a significant increase in risk of asthma compared to patients without epilepsy” or “risk of asthma was not significantly different in patients with epilepsy vs. those without epilepsy”. The same comment applies to line 260 of the discussion and the first sentence of the conclusion in lines 365-367. In the conclusion, “high risk” and “low risk” are used without specifying the referent group.
3. Please add a footnote to Tables 1 and 3 to indicate what the p-values represent and how they were calculated.
4. Clarify footnote a in tables 2 and 4 to indicate that the incidence is per 1000 person-years
5. There were highly significant differences in income between your cases and non-cases in both analyses, but you did not control for income. Was income determined to not be a confounding variable? The assessment of confounding and determination of confounding variables should be discussed in the manuscript.
6. In the discussion or conclusion please discuss impact of these results.

Additional comments

Interesting analysis and well-written paper

·

Basic reporting

The manuscript is well written. The study is based in taiwan however the authors cite prevalence and disease burden rates for the United States. In nature with the international scope of the journal and the nature of this study, it would be better if the author provide some additional background on the global burden of disease as well as some perspective on the prevalence within the study region. The structure fits well with the scope of the article. The authors provide adequate information within the tables in the manuscript and the supplementary content to infer their results.

Experimental design

The primary research is within the scope of the journal. The research question is well defined and fills a clear knowledge gap.The authors tap into a rich data source to adequately address their research question. Taiwan's NHI program provides a longitudinal dataset that allows a retrospective cohort study to evaluate incidence of asthma and epilepsy in patients with pre-existing epilepsy and asthma respectively.

Validity of the findings

The authors found an increased incidence of epilepsy in all asthmatic patients but an increased incidence of asthma only for age groups of 0-10 years and 31-40 years. The authors provide a discussion of plausible biological pathways that may lead to such findings.
The authors adjust for age, sex , region and income in their findings. They posit that both diagnoses share common biological origins such as presence of susceptible genetic loci, common inflammatory triggers such as viral and injury associated cytokines. However if this was the case the authors would have found significant associations for both outcomes. The fact that asthmatics are more prone to epilepsy but not vice versa suggests that it is the sequelae of an asthma diagnosis that result in epilepsy rather than the causal pathways for asthma itself. This may be due to anoxia due to frequent asthma attacks or due to side-effects of anti-asthma medication. It is not clear to me whether the authors would be able to address this using the current dataset. I would suggest that this be clarified further in the Discussion and Conclusion sections. I would also suggest that the authors refrain from using the term " bidirectional" in the discussion and conclusion because teh study only provides conclusive evidence for the increased risk of epilepsy in asthmatics.

---

## Round 0.2 · accepted · Accept

Your responses to the reviewers' comments are satisfactory and the paper is now suitable for publication.